# Resveratrol Improves the Digestive Ability and the Intestinal Health of Siberian Sturgeon

**DOI:** 10.3390/ijms231911977

**Published:** 2022-10-09

**Authors:** Shiyong Yang, Wenqiang Xu, Langkun Feng, Chaoyang Zhang, Chaozhan Yan, Jiajin Zhang, Jiansheng Lai, Taiming Yan, Zhi He, Xiaogang Du, Zongjun Du, Wei Luo, Xiaoli Huang, Jiayun Wu, Yunkun Li

**Affiliations:** 1Department of Aquaculture, College of Animal Science & Technology, Sichuan Agricultural University, Chengdu 611130, China; 2Fisheries Institute, Sichuan Academy of Agricultural Sciences, Chengdu 610066, China; 3Department of Engineering and Applied Biology, College of Life Science, Sichuan Agricultural University, Ya’an 625014, China

**Keywords:** resveratrol, Siberian sturgeon, digestive ability, microbiome

## Abstract

The lack of detailed information on nutritional requirement results in limited feeding in Siberian sturgeon. In this study, resveratrol, a versatile natural extract, was supplemented in the daily diet, and the digestive ability and microbiome were evaluated in the duodena and valvular intestines of Siberian sturgeon. The results showed that resveratrol increased the activity of pepsin, α-amylase, and lipase, which was positively associated with an increase in the digestive ability, but it did not influence the final body weight. Resveratrol improved the digestive ability probably by distinctly enhancing intestinal villus height. Microbiome analysis revealed that resveratrol changed the abundance and composition of the microbial community in the intestine, principally in the duodenum. Random forests analysis found that resveratrol significantly downregulated the abundance of potential pathogens (*Citrobacter freundii*, *Vibrio rumoiensis*, and *Brucella melitensis*), suggesting that resveratrol may also improve intestinal health. In summary, our study revealed that resveratrol improved digestive ability and intestinal health, which can contribute to the development of functional feed in Siberian sturgeon.

## 1. Introduction

The Siberian sturgeon (*Acipenser baerii*) is one of the oldest surviving vertebrates on Earth [1]. It has been widely farmed because of its high growth performance, disease resistance, and short reproductive cycle [2,3]. However, the controlled feeding of Siberian sturgeon is receiving increasing attention, since the formulation of any artificial diet requires detailed information on the specific nutritional requirements and digestive physiology of the candidate species [4]. Its intestinal tissue, a main digestive organ, has also evolved an ecosystem where a host of microbes live [5]. Unlike the external environment, the intestine provides a steady living environment that stably supplies nutrients to the microbes [6]. In return, the microbiota provides the intestine with a wide range of metabolites that maintain the normal metabolic and immune barrier function of the intestine [7]. Hence, the intestine and its microbiota are usually considered as a whole and play an important role in maintaining the body’s healthy.

Previous studies have found that stresses, such as temperature, salinity, starvation, and dissolved oxygen, drive adverse effects on the intestinal microbiota and, ultimately, affect the growth performance of fish species [8,9,10,11]. Amongst these, temperature is a universal and significant factor in affecting the health of fish species, as they are poikilothermic vertebrates and particularly vulnerable to changes in environmental temperature. In cobia, high temperature has been found to affect growth performance by decreasing digestion efficiency [12]. Hani et al. found that the elevated temperature induced reduction in the activity of intestinal amylase and trypsin in three-spined stickleback, thereby influencing its digestive processes [13]. High temperature also affected the composition of the intestinal microbiota, which in turn might negatively regulate digestive processes [14]. 

Resveratrol (trans-3,4,5-trihydroxystilbene), an antitoxin secreted by plants, shows versatile therapeutic potential for treating human diseases because of its biological activities, e.g., antioxidant, anti-inflammatory, anti-cancer, and antibacterial [15,16,17,18,19,20]. Resveratrol also plays an important role in modulation of intestinal microbiota. In high-fat mice, resveratrol was reported to reduce obesity by enhancing the abundance of *Bacteroides*, *Blautia*, *Lachnoclostridium*, *Parabacteroides*, and *Ruminiclostridium* in the intestine [21]. In piglets, resveratrol downregulated the abundance of the phyla Actinobacteria and Firmicutes, and within Firmicutes, the genus *Ruminococcaceae UCG-005* to reduce diquat-induced oxidative stress [22]. Studies in Tilapia (*Oreochromis niloticus*) found that resveratrol increased the proportion of beneficial microbial taxa (Acetobacteriaceae and Methylbacteraceae), while the proportion of harmful microbial taxa (Streptococcaceae) was decreased [23]. These studies have shown that resveratrol can improve the health of intestinal microbiota.

Our study aimed to explore the effect of resveratrol on the intestinal health of Siberian sturgeon. Resveratrol was supplied into the daily diet and after 45 days, morphology, digestive ability and microbiome of the duodenum and valvular intestine were evaluated. This study revealed the positive effects of resveratrol on intestinal health, which contributes to the application of resveratrol in aquaculture and provides a reference for the functional development of resveratrol in humans.

## 2. Results

### 2.1. Resveratrol Improved Intestinal Digestion and Protected the Intestinal Structure in Siberian Sturgeon

To visualize the entire experimental design, a concise experimental flow chart was drawn (Figure 1A). During the experiment, the feeding rates of the four groups were tallied daily. The results showed that sturgeon from the LR group showed a significantly higher feeding rate compared to those in group C (*p* < 0.05), while the feeding rate of the HR group showed an increasing trend compared to group C (Figure 1B). However, no significant differences were observed in final body weight (FBW) and percentage weight gain (PWG) among the groups (Table 1).

The duodenum and valvular fragments of the intestinal tissue were sampled to observe morphological and histological changes. The valvular intestinal contents were significantly less in the LR and HR groups than in group C (Figure 1C). Histological observations showed a significant increase in the height of intestinal villus in both the duodenum and valvular intestine of Siberian sturgeon fed with resveratrol in groups LR and HR compared to group C (*p* < 0.05) (Figure 1D–G). Additionally, in the valvular intestine, the muscular layer thickness in the LR group was markedly increased compared to that of the C group (*p* < 0.05), but no significance was observed in the HR group (Figure 1G).

### 2.2. Resveratrol Increased Digestive Enzyme Activity of Siberian Sturgeon

To further explore the effects of resveratrol on the intestinal digestion capacity of Siberian sturgeon, the activity of digestive enzymes (pepsin, α-amylase, lipase) was detected in the duodenum and valvular fragments. As shown in Figure 2A–C, in the duodenum, the activity of all three digestive enzymes tended to decline with the increase of farming time (C group vs. BB group), while the activity of three digestive enzymes was enhanced in the LR and HR groups compared to that of group C, especially in the LR group that showed statistical significance (*p* < 0.01). This result was also found in pepsin activity of the valvular intestine (Figure 2D). Of α-amylase and lipase activity in the valvular intestine, no significant change was observed between BB and C groups, despite α-amylase activity in the C group being higher than in the BB group (Figure 2E,F). In the valvular intestine, the α-amylase and lipase activities were also increased in the LR and HR groups relative to the C group, but only lipase activity in the HR group showed significance (*p* < 0.05, Figure 2E,F).

### 2.3. Global Analysis of OTUs

To investigate the influence of resveratrol on the gut microbiota of Siberian sturgeon, 24 samples from the duodenum and valvular intestine of Siberian sturgeon were analyzed (Appendix A). A total of 600,418 valid sequences were obtained with an average length of 1,413 bp (Appendix A). A total of 1,656 OTUs were classified at the level of species (73%), genus (89%), family (94%), order (97%), class (99%), and phylum (99%). We assessed the diversity and depth of microbial community structure. The Shannon’s index and Simpson’s index were used to compare community diversity among samples. No significant difference (*p* > 0.05) was found in the diversity among the samples of the duodenum or valvular intestine (Appendix A). The community richness was measured by Chao index, which indicated no difference (*p* > 0.05) among samples of the duodenum or valvular intestine (Appendix A). The coverage index showed that the community coverage of all samples was more than 99% (Appendix A). Rank-abundance and dilution curves that illustrate the difference in community richness and evenness among the samples, respectively, did not show significant difference (Appendix A). The raw 16S rRNA sequencing data have been submitted to the Sequence Read Archive (SRA) database of the National Center for Biotechnology Information (NCBI), with accession number PRJNA863445.

Beta diversity analysis showed that the variability of microbial communities in the duodenum samples was higher than in the valvular intestine samples (Figure 3). The microbial community was dominated by Pseudomonadaceae in the valvular intestine samples, and the abundance of other families accounted for a relatively low percentage. In the duodenum samples, Pseudomonadaceae was also the dominant community, but unlike the valvular intestine, other communities, such as Enterobacteriaceae and Rhodobacteraceae, also showed higher abundance. Principal coordinate analysis revealed marked differences in the microbial community structure of different groups, particularly in the valvular intestine (Appendix A).

### 2.4. Resveratrol Alters the Community Composition in the Duodenum of Siberian Sturgeon

OTU levels among different groups were analyzed to compare the species distinctiveness (Figure 4). The Venn annotation showed that the community composition of both the duodenum and valvular intestine presented distinct diversity among different groups. In the duodenum samples, a total of 895 OTUs were identified, of which only 224 OTUs were shared in BB, C, LR and HR groups (Figure 4A). A total of 739 OTUs were identified in the valvular intestine, but only 183 OTUs were shared in the BB, C, LR, and HR groups (Figure 4B). Additionally, in both the duodenum and valvular intestine, the number of microbial species was decreased significantly with the increasing of culturing time (BB group vs. C group), while diets supplemented with different doses of resveratrol increased the number of microbial species, in comparison to the C group (Figure 4C,D).

In the community composition chart, the microbiota of Siberian sturgeon in the duodenum and valvular intestine mainly belonged to five phyla: Proteobacteria, Fusobacteria, Firmicutes, Actinobacteria, and Bacteroidetes (Appendix A). The Proteobacteria occupied the largest proportion in all groups. In the duodenum of the HR group, the ratio of Fusobacteria was greater than in other groups, but this event did not happen in the valvular intestine (Appendix A). Further analysis revealed that *Fusobacterium* was identified as the differential microbiota in the duodenum between the HR group and the C group, while *Escherichia coli* was the differential microbiota in the duodenum of LR group (Appendix AE). However, marked diversity was found in the microbiota composition among the groups in the valvular intestine (Appendix A).

### 2.5. Resveratrol Influenced the Abundance of Duodenum Microbiota of Siberian Sturgeon

The community abundance of the BB/LR/HR groups vs. the C group was analyzed in the duodenum and valvular intestine (Figure 5). In the duodenum, the abundance of Sphingomonas hengshuiensis, Sphingomonas sp. LK11 in the BB group was significantly decreased compared to that in the C group, but no community with the increased abundance was found. In the LR vs. C group, the abundance of Escherichia coli was significantly increased, while the abundance of Salmonella enterica and unclassified Enterobacterriaceae was observably decreased. In the HR vs. C group, the communities with the significantly increased abundance were unclassified Rhodobacter, Rhodopseudomonas palustris, and Ilyobacter polytropus, while the abundance of unclassified Fusobacterium was markedly decreased. By contrast, only unclassified Rhizobiaceae in the LR vs. C group showed significant increase in the community abundance in the valvular intestine.

### 2.6. Resveratrol Reduced the Abundance of Potential Pathogens in the Duodenum of Siberian Sturgeon

A cladogram of LEfSe was constructed to find the most crucial microbiota (microbial biomarkers) in the duodenum among different groups (Figure 6A). The most dominant microbe was the following families: Lactobacillaceae in the BB group; Sphingomonadaceae, Vibrionaceae, and Xanthomonadaceae in group C; Helicobacteraceae and Enterobacteriaceae in the LR group; and Fusobacteriaceae in the HR group. Random forest analysis was carried out to determine variable importance of the samples from the BB, C, LR, and HR groups (Figure 6B). The results indicated that the potential pathogens (*Citrobacter freundii*, *Vibrio rumoiensis*, and *Brucella melitensis*) were the critical bacterial species in all groups, and their abundances were markedly reduced in the LR and HR groups compared to the C group (*p* < 0.05).

## 3. Discussion

Resveratrol has been reported to function as a regulator for the intestinal microbial community and as a protector of the integrity of the intestinal barrier [24,25,26], protecting mainly through its anti-inflammatory modulation [27,28,29]. In aquatic animals [30], resveratrol improved the absorption of nutrients in the intestinal tissues of zebrafish by increasing the number of goblet cells as well as the width and height of the villi. This study showed that diet supplemented with resveratrol increased the feeding rate of Siberian sturgeon but did not influence the final body weight. Generally, increase in food intake directly relates body- weight gain, even obesity [31,32]. By contrast, one study reported that diet supplemented with resveratrol lessened the body mass and whole-body fat in blunt snout bream, highlighting the hypolipidemic effect of resveratrol [33]. Thus, we speculated that resveratrol, being a versatile dietary additive, not only improves the nutrient uptake but also prevents obesity in fish. 

Digestive capacity of the intestine is critical for energy absorption [34], however, we currently have limited knowledge about the digestive capacity of the intestine in aquaculture species, especially in cold-water species. Digestive enzymes directly reflect the digestive capacity, and in this study, the activity of pepsin, α-amylase, and lipase was assayed in the duodenum and valvular intestine. The results showed that supplementing the diet with resveratrol significantly increased the activity of pepsin, α-amylase, and lipase in both the duodenum and valvular intestine of Siberian sturgeon. This result indicates that resveratrol supplementation improved digestive capacity by enhancing the activity of digestive enzymes, which led to the increased feeding rate (Figure 1B). Tagliazucchi et al., found that resveratrol could bind pepsin and change its three-dimensional conformation to make it more active in pork [35]; this may be the reason why resveratrol increased pepsin activity in this study, while the reason for the ability of resveratrol to increase α-amylase and lipase activity remains to be further investigated.

The intestinal lining, which is in direct contact with the external environment, may become a major site for colonization and growth of microbiota [36,37]. In aquaculture species, intestinal microbiota is of significance to maintaining homeostasis [38,39]. In this study, the composition and abundance of microbial communities were evaluated and showed distinct diversity among different groups (Figure 3, Figure 4 and Figure 5). More specifically, supplementing the diet with resveratrol significantly influenced the abundance and composition of the bacterial community in the duodenum of Siberian sturgeon. After feeding with resveratrol, the community abundance of *unclassified Rhodobacter*, *Rhodopseudomonas palustris*, *Ilyobacter polytropus*, and *Escherichia coli* were significantly elevated, while the community abundance of *unclassified Fusobacterium*, *Salmonella enterica*, and *unclassified Enterobacteriaceae* were distinctly reduced in the duodenum (Figure 5). Among the microbiota of significantly elevated abundance, *unclassified Rhodobacter* was not annotated to any species, but its members can exert hypocholesterolemic effects by assimilating cholesterol into bile and producing bile acids [40]. *Rhodopseudomonas palustris* has been applied as one of the potential probiotics in the feed industry due to its extraordinary metabolic versatility [41]. *Ilyobacter polytropus* may be related to the degradation of monomeric 3-hydroxybutyrate, thus promoting the utilization of butyrate in the organism [42]. Among the significantly reduced components of the microbiota, the genus *Fusobacterium* has been proven to be a potentially pathogenic bacterium and implicated in human colorectal cancer [43,44]; *Salmonella enterica* is a well-versed pathogenic that can cause death in humans and animals through intestinal infection [45,46]; and *unclassified Enterobacteriaceae* was not mapped to any species, but *Enterobacteriaceae* has also received much concern for its extensive drug-resistance [47]. Therefore, our study revealed that resveratrol increased several intestinal probiotic bacteria (*unclassified Rhodobacter, Rhodopseudomonas palustris,* and *Ilyobacter polytropus*) and decreased several potentially pathogenic bacteria (*Fusobacterium*, *Salmonella enterica*, etc.), and was thereby associated with positive regulation of the microbial community in the duodenum of Siberian sturgeon.

In this study, the LEfSe algorithm and random forest analysis were constructed to determine microbial markers of each group and variable importance of all the samples, respectively, in the duodenum. The results revealed that the *Citrobacter freundii*, *Vibrio rumoiensis,* and *Brucella melitensis* were the most important bacterial species (could distinguish resveratrol-feeding groups from control group with the highest accuracy), and their abundance was significantly decreased in the duodenum after feeding with resveratrol (Figure 6B). These components of the microbiota were identified as the potential pathogens [48,49,50], indicating that resveratrol could reduce the composition of the pathogen community, consistent with previous studies [51,52]. The valvular intestine is a special structure of sturgeon, alongside the duodenum, to digest and absorb nutrients from food [53], but no significant influence of resveratrol was found on the valvular intestine microbiota of Siberian sturgeon. Thus, resveratrol may specifically influence the microbial community of the duodenum, but the mechanism is needed to be investigated in the future.

## 4. Materials and Methods

### 4.1. Fish and Experimental Design

A total of 300 fish (10-month-old, 248.1 ± 5.9 g) were purchased from a sturgeon farm (Tianquan County Chuanze Fishery Co., Ltd., Ya’an, China) and randomly assigned to circular plastic tanks (1.5 m in diameter and 1 m in height) for one week of acclimatization. The sturgeons were reared in flowing water, the water temperature, dissolved oxygen, pH, ammonia nitrogen, and nitrite were maintained at 16.0 ± 0.5 °C, 8.0 ± 0.6 mg/L, 7.6 ± 0.2, ≤ 0.01 mg/L, and ≤ 0.05 mg/L, respectively. The sturgeon were fed with commercial feed (Haida Group Co., Ltd., Guangdong, China) 3 times per day (at 8:00 am, 14:00 pm, and 20:00 pm, respectively), each fed 1% of the total fish weight, and after 1 h, the residual feed was collected and weighed after drying to calculate the feeding rate. Feeding rate (%) = (1 − average residual feed/average feeding feed) × 100; average residual feed (g/fish/day) = [$\sum_{k=1}^{n}\left(\mathrm{residual}\;\mathrm{feed}\;\mathrm{per}\;\mathrm{day}/\mathrm{fish}\;\mathrm{number}\right)$*_k_*]/*n*, *n* = day number of each group; average feeding feed (g/fish/day) = [$\sum_{k=1}^{n}\left(\mathrm{feeding}\;\mathrm{feed}\;\mathrm{per}\;\mathrm{day}/\mathrm{fish}\;\mathrm{number}\right)$*_k_*]/*n*, *n* = day number of each group. The fish number was dynamically counted based on the actual survival number. 

After the acclimatization, 30 fishes (divided into 3 replicates) were randomly selected from the 300 fish and set as the before breeding (BB) group, while the rest of the fish were randomly assigned to 3 groups: control (C) group, fed with commercial feed; LR group, diet supplemented with low-dose resveratrol (0.16 mg/kg); HR group, diet supplemented with high-dose resveratrol (0.32 mg/kg). Each group contained 3 tanks, 30 fishes per tank (Figure 1A). Resveratrol (purity ≥ 99%) was purchased from Macklin Biochemical Co., Ltd (Shanghai, China). All animal handling procedures were approved by the Animal Care and Use Committee of Sichuan Agricultural University, following the guidelines of animal experiments of Sichuan Agricultural University under permit number 015-01521300.

### 4.2. Sample Collection, DNA Isolation, Library Construction, and Sequencing

At the end of the feeding, the duodena and valvular intestines of nine fish from each group were collected and fixed in 10% neutral formalin buffer for at least 24 h for histological observation. For microbiome analysis, nine fish were randomly selected from each group and anesthetized using 200 mg/L MS-222 (Jinjiang Aquatic Supplies Co., Ltd., Fujian, China). Intestinal tissues, containing duodena and valvular fragments, were collected under aseptic condition and then stored at −80 °C. Genomic DNA was extracted from the intestinal bacteria using a bacterial DNA isolation kit (Foregene, Chengdu, China). After the extraction, DNA fragments from the samples were amplified using specific primers: 8F: 5′-AGAGTTTGATCATGGCTCAG-3′, 1492R: 5′-CGGTTACCTTGTTACGACTT-3′. The PCR products were detected by electrophoresis in a 1% agarose gel, purified using a Zymoclean Gel Recovery Kit (D4008), quantified by the QuantiFluorTM-ST Blue Quantitative Fluorescence System (Promega, Beijing, China), and then subjected to next-generation sequencing. V3-V4 amplicon libraries were constructed and pair-end (PE) sequencing was performed on the MiSeq system (Illumina, San Diego, CA, USA). Based on the overlapping PE reads, pairs of reads were merged into sequences using Flash software (version: 1.2.11, http://ccb.jhu.edu/software/FLASH/index.shtml (accessed on 9 August 2021)). To further analyse, the following sequences were excluded from the dataset: (i) shorter than 1400 bp, (ii) greater than 1600 bp in the library, or (iii) contained 10 consecutive identical bases.

### 4.3. Taxonomy Classification and Differential Abundance Analysis

The acquired sequences were clustered into operational taxonomic units (OTUs) using UPARSE (http://drive5.com/uparse (accessed on 15 August 2021)). Each OTU was compared with the 16s rRNA database (Silva) to obtain classification information of the species. The RDP Classifier Bayes algorithm was used to classify and analyze the OTU representative sequences with 97% similarity, and the corresponding sequences were assigned to each classification level: kingdom, phylum, class, order, family, genus, and species. In addition, alpha diversity, including Shannon, Simpson, Chao and Coverage indices were calculated by Mothur. Hierarchical clustering analysis was performed based on the beta diversity distance matrix, and the UPGMA algorithm was used to construct clustering trees to compare the similarity of community composition. Species with relative abundance less than 0.01 in all samples were categorized as “other”. Differential species analysis was performed by LEfse tool (https://bitbucket.org/biobakery/biobakery/wiki/Home (accessed on 18 August 2021)), random forest analysis was carried out with the randomForest package in R studio, and community function prediction was conducted based on the functional gene profile of microorganisms in the KEGG database.

### 4.4. Histological Observation

The fixed intestinal tissues were dehydrated, transparentized with xylene, and embedded in paraffin wax. The solidified wax blocks were cut into 5-mm slices and mounted on slides for hematoxylin and eosin (H&E) staining. After staining, the slides were observed under optical microscope (Nikon, Tokyo, Japan). To evaluate the effects of resveratrol on the intestine of Siberian sturgeon, the intestinal parameters (mucosal thickness, muscular layer thickness, and villus height) were assessed with reference to the previous study. The thicknesses of the mucosal layer and muscular layer were defined as the average of five measurements per fish. The villus height was measured from the tip of the villus to the base, five measurements per sample.

### 4.5. Digestive Enzyme Activity Assay of Intestine

The duodenum and valvular fragments of the intestinal tissue were ground with buffer solution using commercial kits (A080-1-1, C016-1-1, A054-1-1, Nanjing Jiancheng Bioengineering Institute, Nanjing, China). The activity of pepsin, α-amylase, and lipase was assayed with the above-mentioned commercial kits. All operations were performed in strict accordance with the manufacturer’s description.

### 4.6. Statistical Analysis

Data were expressed as mean ± SD. Statistical analysis was performed using one-way ANOVA in SPSS version 27.0 software. Least Significant Difference (LSD) was used to analyze the differences between BB/LR/HR and C groups. Principal coordinates analysis was used to cluster different samples according to explanatory parameters. The *p*-value less than 0.05 was considered as the statistical significance.

## 5. Conclusions

In conclusion, the elevated activity of digestive enzymes and the increased height of intestinal villi were found after feeding with resveratrol in Siberian sturgeon. The function of resveratrol to inhibit the abundance of potentially pathogenic bacteria in the duodenum was also revealed. These results indicate that resveratrol may improve intestinal digestion and protect the microbial barrier. The information presented in the present work will be helpful to further the study of the intestinal function of Siberian sturgeon and expand the application of resveratrol in aquaculture.

## Figures and Tables

**Figure 1 ijms-23-11977-f001:**
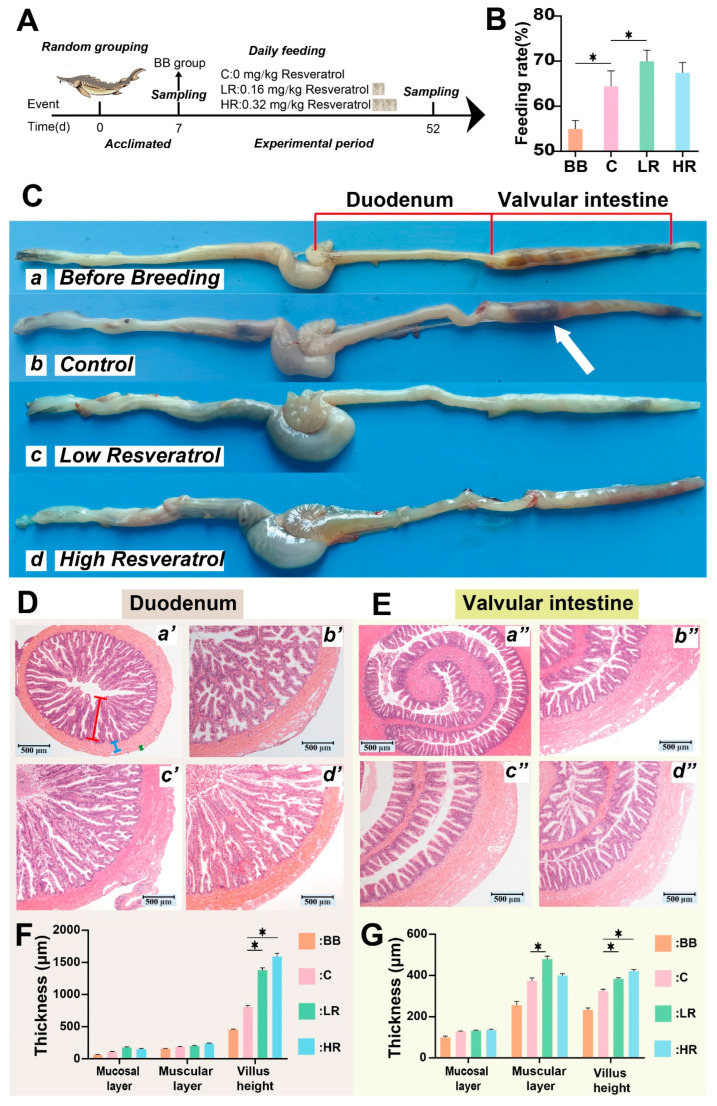
**Experimental flowchart, feeding rate, and histological observation of****Siberian sturgeon.** (**A**) Experimental flowchart, sampling at day 7 indicates before breeding (BB) group, sampling at day 52 includes three groups: control (C), low resveratrol (LR), and high resveratrol (HR). (**B**) Feeding rate (%) = (1 − average residual feed/average feeding amount) × 100. (**C**) Representation of intestinal tissues from different groups. Intact tissues, containing duodenum and valvular intestine fragments, were sampled from each group at the termination of the feeding administration. The white arrow indicates intestinal contents. (**D**,**E**) Histological changes in the duodenum and valvular intestine. Sampled intestinal tissues from B were divided into duodenum and valvular intestine fragments that underwent H&E staining to observe histological change (n = 3). Representation of histological change is shown as follows: BB (**a’** and **a’’**), C (**b’** and **b’’**), LR (**c’** and **c’’**), and HR (**d’** and **d’’**). Red scale bars indicate length of intestinal villus; blue scale bars represent muscular layer thickness; and green scale bars indicate mucosal layer. (**F**,**G**) Thickness of mucosal layer, muscular layer, and intestinal villus length in different groups. * *p* < 0.05 vs. the control group; two-way ANOVA plus Bonferroni post-tests.

**Figure 2 ijms-23-11977-f002:**
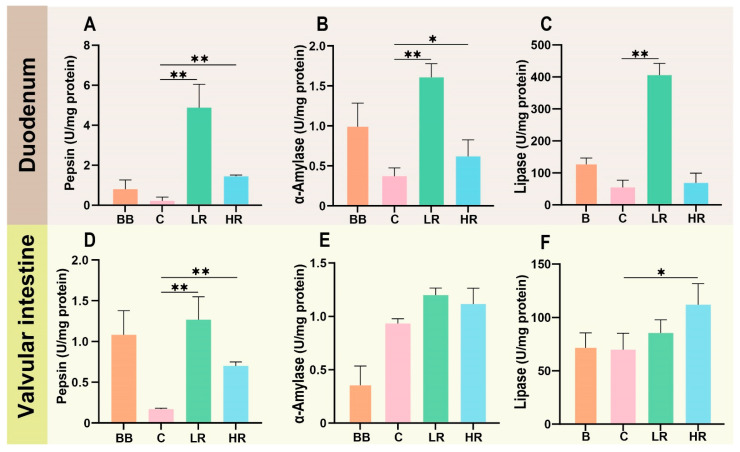
**Effects of resveratrol on the activity of intestinal digestive enzymes.** (**A**–**C**) Pepsin, α-amylase, lipase activity in the duodenum. (**D**–**F**) Pepsin, α-amylase, lipase activity in the valvular intestine. * *p* < 0.05, ** *p* < 0.01 vs. the control group; one-way ANOVA plus Bonferroni post-tests.

**Figure 3 ijms-23-11977-f003:**
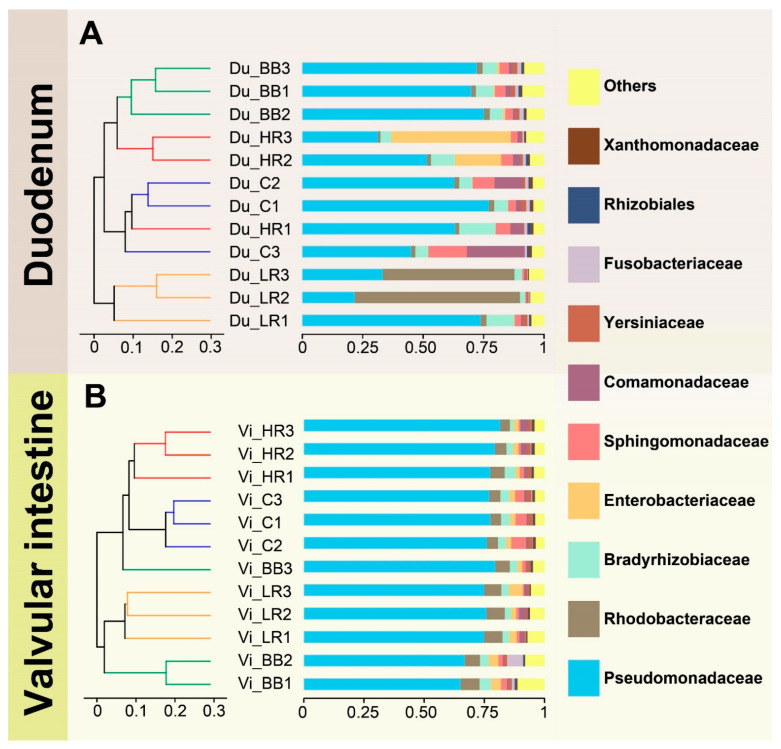
(**A**,**B**) represent hierarchical clustering trees of Siberian sturgeon duodenum and valvular families, respectively. Clustering analysis based on Bray–Curtis dissimilarity.

**Figure 4 ijms-23-11977-f004:**
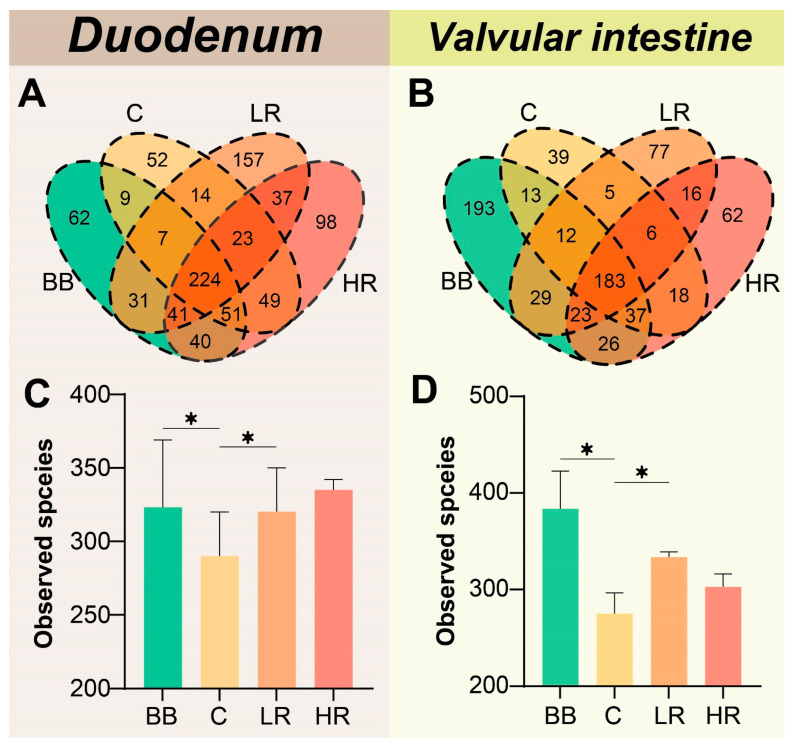
**Community composition analysis among different groups.** (**A**,**B**) Venn diagrams of OTU numbers in different groups. (**C**,**D**) Statistics of species number among different groups, * *p* < 0.05 vs. the control group; one-way ANOVA plus Bonferroni post-tests.

**Figure 5 ijms-23-11977-f005:**
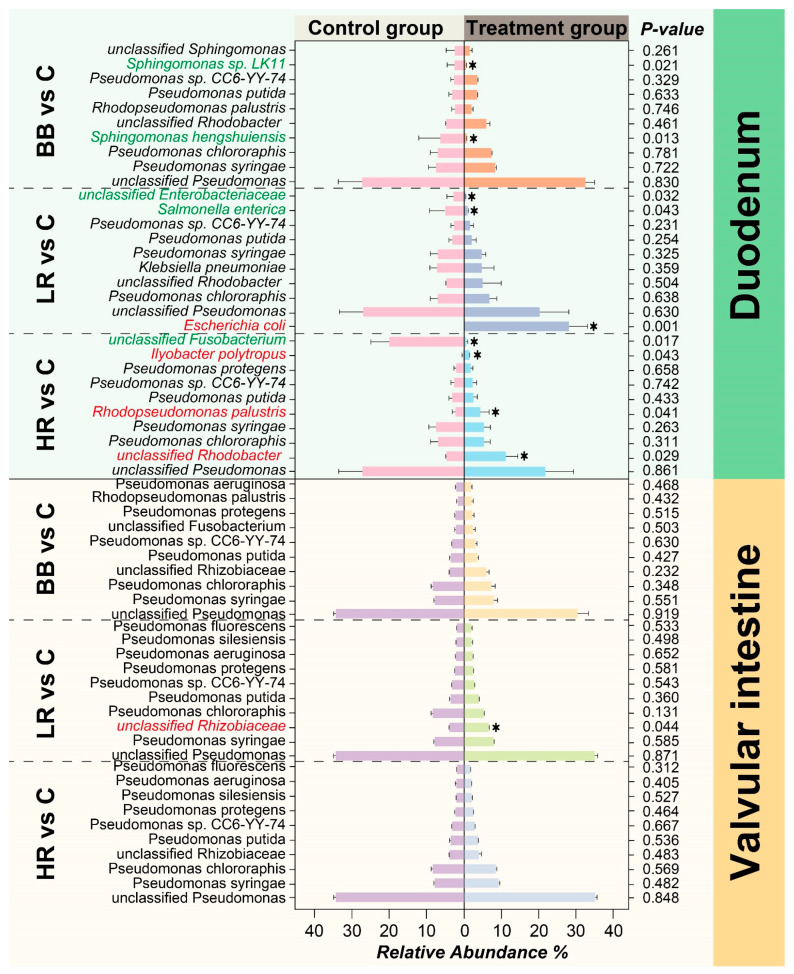
**Comparison of microbial community between the treatment and control groups.** The chart lists the top 10 abundant communities in the duodenum and valvular intestine in BB vs. C, LR vs. C, and HR vs. C groups. The data are presented as means ± standard deviation, Wilcoxon rank-sum test was applied to determine if differences between groups were significant (* *p* < 0.05). The communities with red and green fonts indicate significant increase and decrease in abundance, respectively.

**Figure 6 ijms-23-11977-f006:**
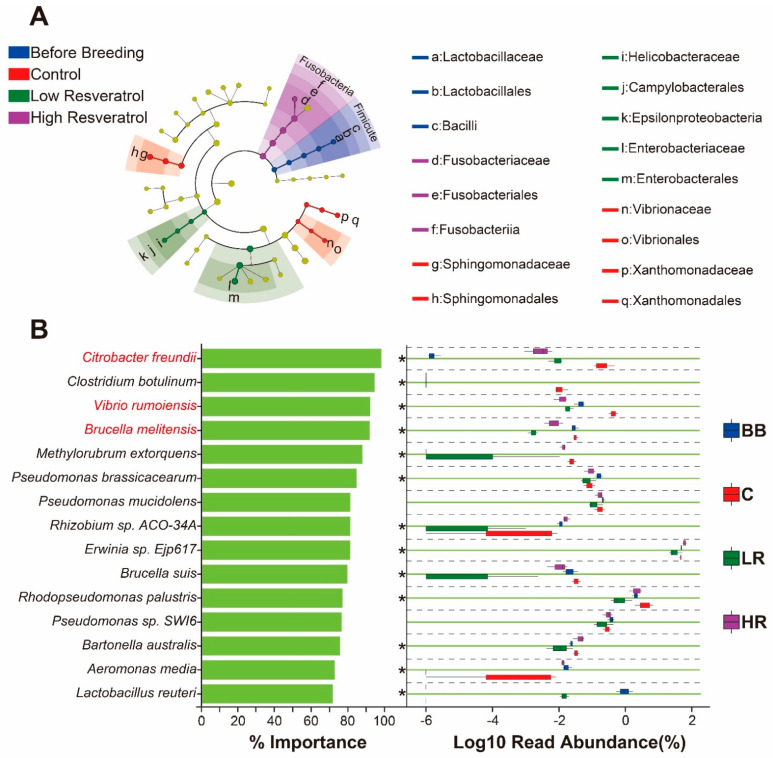
Identification of microbial markers for different groups by LEfSe analysis and random forest analysis for the samples from all groups in the duodenum. (**A**) Taxonomic cladogram generated from LEfSe analysis. Taxonomic characterization of differences in intestinal tissues among different groups; concentric circles from outside to inside indicate different taxonomic classes (phylum to family), and different color areas point to different groups. The abundance of intestinal microbiota was revealed by the size of each node. (**B**) Variable importance plot from random forest classifier. Left graph shows top 15 bacteria with % importance (mean decrease in Gini) ≥ 70 among different groups, bacteria marked in red font indicate potential pathogenic bacteria. Right figure represents abundance of corresponding communities by boxplots in different groups, with “*” on left side representing the difference significance of BB/LR/HR compared to C (*p* < 0.05).

**Table 1 ijms-23-11977-t001:** The effect of resveratrol on the growth performance of Siberian sturgeon.

Sample Category	Replicate	IBW (g/fish)	FBW (g/fish)	PWG (%)
Before breeding	BB1	248.3 ± 2.6 ^a^	/	/
	BB2	249.8 ± 5.3 ^a^	/	/
	BB3	251.2 ± 4.3 ^a^	/	/
Control	C1	247.9 ± 5.1 ^a^	357.6 ± 8.6 ^b^	44.53 ^c^
	C2	250.5 ± 2.9 ^a^	364.8 ± 7.0 ^b^	45.63 ^c^
	C3	249.9 ± 6.5 ^a^	371.0 ± 15.6 ^b^	48.46 ^c^
Low resveratrol	LR1	253.2 ± 3.8 ^a^	372.4 ± 13.5 ^b^	47.08 ^c^
	LR2	252.3 ± 4.3 ^a^	369.1 ± 9.5 ^b^	46.29 ^c^
	LR3	249.4 ± 1.6 ^a^	360.5 ± 6.3 ^b^	44.55 ^c^
High resveratrol	HR1	255.1 ± 5.6 ^a^	378.5 ± 10.1 ^b^	48.37 ^c^
	HR2	249.7 ± 5.8 ^a^	374.8 ± 13.0 ^b^	50.01 ^c^
	HR3	251.9 ± 6.5 ^a^	366.1 ± 6.5 ^b^	45.34 ^c^

Data are expressed as mean ± SD, and the different letters in the same column mean the significance (*p* < 0.05). IBW: initial body weight; FBW: final body weight; PWG: percent weight gain = (FBW − IBW)/IBW.

## Data Availability

The raw 16S rRNA sequencing data have been submitted to the Sequence Read Archive (SRA) database of the National Center for Biotechnology Information (NCBI) with accession number PRJNA863445.

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
