# Peer review of "Resveratrol Improves the Digestive Ability and the Intestinal Health of Siberian Sturgeon"

_ijms, 2022, doi:10.3390/ijms231911977_

Round 1

Reviewer 1 Report

Title: Resveratrol improves the digestive ability and the intestinal health of
Siberian sturgeon
Authors: Shiyong Yang, Wenqiang Xu, Langkun Feng, Chaoyang Zhang,
Chaozhan Yan, Jiajin Zhang, Jiansheng Lai, Taiming Yan, Zhi He, Xiaogang
Du, Zongjun Du, Wei Luo, Xiaoli Huang, Jiayun Wu, Yunkun Li

Summary:

The authors have investigated a relevant topic in an interesting way. The work is structured in a comprehensible way, the figures are clearly presented. In some places, details could be added or explained to increase the significance of the work and avoid misunderstandings.

Examples: 

1) Introduction

a) Line 54 "Anti-cancer". Since the focus of the paper is on the digestive tract, the effect of resveratrol against colorectal cancer (CRC) should be much more emphasized here.

Please add an additional reference:

doi: 10.3390/ijms23094714

doi: 10.3390/cancers14061372.

doi: 10.1007/s12032-021-01611-w.

doi: 10.3389/fphar.2022.978625.

b) Lines 64-68: It is not clear to me throughout the paper whether the results of the study are intended specifically to improve fish culture or whether subsequent transferability to humans is intended. This could possibly be clarified at this point. 

2) Results

a) Line 75/Figure 1B/Line 223: Please define the term "feeding rate." Since the fish were fed three times a day according to M&M, I can't imagine what is meant by this and what the percentages mean.

b) Lines 79-84: Please describe in more detail what is shown in the pictures in Figure 1C/D/E.

c) Figure 5+6: Please add to the legends what the bacteria written in green/red mean. 

3) Discussion

a) Line 220: Please add that resveratrol protects mainly through its anti-inflammatory modulation.

Please add an additional reference:

doi: 10.3390/molecules25184292

doi: 10.1177/09603271211041678.

doi: 10.3390/jcm9061796.

b) Line 235: Have these enzyme changes been found in other species? Please add a reference if appropriate.

c) Line 269: For what are these bacterial species the "most important"? Please specify.

d) Line 273-277: The text is misleading. It sounds like resveratrol only affects the "unimportant" part of the intestine. Therefore, I recommend emphasizing the importance of the duodenum here as well. 

4) Material and methods

a) line 280: Please indicate the age of the fish purchased. Were they full grown?

Reviewer 2 Report

In my opinion the manuscript is novel and interesting. The subject is important in the context of aquaculture nutrition as well as molecular sciences. The fish species is very well selected.

The Authors must add basic water parameters to the manuscript because some of them are lacking. Without given concentration of ammonia (NH3), nitrites (NO2-) and nitrates (NO3-) or at least detailed explanation why they were not measured included in the manuscript, this paper should not be accepted. Moreover, MS 222 concentration used for fish euthanasia must be supplemented.

I have no other comments on methodology. Results and Discussion are well written. Abstract is informative however names of fish pathogens (Citrobacter freundii, Vibrio rumoiensis and Brucella melitensis) should be written in italics.

The strength of the paper is very good way of the presentation of the Results obtained. Key words are correct. Conclusion is correct.

My recommendation: Minor revision
